# Prediction of Impulsive Aggression Based on Video Images

**DOI:** 10.3390/bioengineering10080942

**Published:** 2023-08-08

**Authors:** Borui Zhang, Liquan Dong, Lingqin Kong, Ming Liu, Yuejin Zhao, Mei Hui, Xuhong Chu

**Affiliations:** 1School of Optics and Photonics, Beijing Institute of Technology, Beijing 100081, China; 2Beijing Key Laboratory for Precision Optoelectronic Measurement Instrument and Technology, Beijing 100081, China; 3Yangtze Delta Region Academy of Beijing Institute of Technology, Jiaxing 314019, China

**Keywords:** impulsive aggression, imaging photoplethysmography technology, heart rate variability, facial expression

## Abstract

In response to the subjectivity, low accuracy, and high concealment of existing attack behavior prediction methods, a video-based impulsive aggression prediction method that integrates physiological parameters and facial expression information is proposed. This method uses imaging equipment to capture video and facial expression information containing the subject’s face and uses imaging photoplethysmography (IPPG) technology to obtain the subject’s heart rate variability parameters. Meanwhile, the ResNet-34 expression recognition model was constructed to obtain the subject’s facial expression information. Based on the random forest classification model, the physiological parameters and facial expression information obtained are used to predict individual impulsive aggression. Finally, an impulsive aggression induction experiment was designed to verify the method. The experimental results show that the accuracy of this method for predicting the presence or absence of impulsive aggression was 89.39%. This method proves the feasibility of applying physiological parameters and facial expression information to predict impulsive aggression. This article has important theoretical and practical value for exploring new impulsive aggression prediction methods. It also has significance in safety monitoring in special and public places such as prisons and rehabilitation centers.

## 1. Introduction

Impulsive aggression is a term used to express the tendency of individuals to act aggressively without careful consideration [1,2]. Compared with normal individuals, special individuals (such as those who are compelled to undergo treatment and those who are imprisoned) have higher impulsiveness [3]. They are prone to quick and unplanned reactions, ignoring the negative consequences of these reactions in response to internal or external stimuli. This behavioral tendency poses a threat to site management and personnel safety. Therefore, it is very important and practically significant to screen and predict special individuals with impulsive aggression in advance, reducing the possibility of impulsive aggressive behavior through intervention and treatment.

However, the existing research methods for assessing impulsive aggression primarily rely on questionnaires. Questionnaires are dependent on the subjective perceptions and judgements of the subjects concerning their own feelings. As each individual may have varying perceptions and experiences regarding the same issues, the measurement outcomes tend to be subjective. While behavioral and neurological measures offer a more objective approach, they cannot fully replace questionnaire measures due to their demanding requirements for measurement environments and equipment.

At the same time, multiple studies have shown that there is a significant correlation between impulsive aggression and negative emotions, particularly anger, anxiety, and other negative emotions [4,5,6]. Moreover, studies have also found that individuals often exhibit intensified negative emotions when engaging in aggressive behavior [7]. Thus, it is possible to predict impulsive aggression by detecting an individual’s negative emotions [8].

Heart rate variability (HRV) refers to fluctuations in the time interval between consecutive heartbeats and is usually used to reflect the functional activity of the autonomic nervous system [9]. Previous studies have shown that when individuals experience negative emotions, their sympathetic nervous system activity increases, their respiratory rate accelerates, their heart rate (HR) significantly increases, the high frequency (HF) of the power spectrum of heart rate variability decreases, and the ratio of low frequency to high frequency (LF/HF) increases, especially when experiencing emotions such as anger and anxiety [9]. Conversely, when individuals experience positive emotions, such as happiness, their RR intervals become longer, their heart rate slows down, and the low-frequency power (LF) of their heart rate variability power spectrum also decreases [9]. The regulation of heart rate variability is associated with the functional activities of the sympathetic and parasympathetic nervous systems, where the high-frequency component is mainly controlled by the parasympathetic nervous system, and the low-frequency component is regulated by the sympathetic nervous system [10]. Thus, by analyzing the time-domain and frequency-domain parameters of heart rate variability, it is possible to analyze an individual’s current autonomic nervous system state and further analyze their emotional state [11]. Facial expressions can also reflect an individual’s current emotional and psychological states [12]. Ekman and Friesen defined the six culturally universal facial expressions that represent six basic emotions (happiness, disgust, surprise, sadness, anger, and fear) in the 1970s [12]. When experiencing negative emotions, individuals typically express emotions such as disgust, sadness, anger, and fear, while when experiencing positive emotions, they express emotions such as happiness and surprise [13]. As impulsive aggression is closely related to an individual’s emotional state, analyzing an individual’s facial expressions to predict and prevent impulsive aggression is possible.

Therefore, this study breaks through the traditional questionnaire survey method and proposes an impulse attack behavior prediction method that integrates physiological parameters and facial expression information. The method uses video to extract the subject’s heart rate variability and facial expression information through IPPG technology and facial recognition technology, then predicts impulsive aggression by fusing heart rate variability characteristic parameters with facial expression information. This method is non-contact and non-interfering and can realize the early screening and prediction of impulsive aggression tendencies in special individuals. It is more objective than questionnaires and has important significance for exploring new impulsive aggression prediction methods.

The remainder of this paper is organized as follows: In Section 2, we provide an overview of related works in the field. Section 3 outlines our proposed video-based impulse attack behavior prediction method, providing comprehensive and detailed descriptions. The experimental setup and introduction are presented in Section 4, followed by a display of the experimental results in Section 5. Finally, in Section 6, we present our conclusion, summarizing the key findings and contributions of this study.

## 2. Related Works

In recent years, numerous scholars have conducted research on the psychological status of special populations. For instance, Maria et al. conducted predictive research on the trait aggression of recidivists [14], while Ricarte et al. studied suicidal behavior of incarcerated males [15]. These studies utilized the Buss–Perry aggression questionnaire and impulsive premeditated aggression scale to assess whether the subjects exhibited impulsive aggression.

Additionally, Hausam et al. scored and classified prisoners’ aggression based on information at the time of imprisonment and observations of their behavior through prison guards during the initial weeks of incarceration [16]. Seid et al. measured antisocial personality disorder (ASPD) through face-to-face interviews and *The Diagnostic and Statistical Manual of Mental Disorders* (DSM-5) [17].

Furthermore, Wang et al. proposed an “implicit-based and explicit secondary ‘implicit + explicit’” screening evaluation system [18] to assess impulsiveness among special individuals. Shi et al. developed the “Rain Man” painting test to evaluate the impulsive aggression of individuals compelled to undergo treatment [19]. This test evaluates psychological pressure and aggressiveness by scoring different details of the drawn pictures.

Overall, the evaluation of the psychological status of special individuals currently relies predominantly on questionnaires, inquiries, and observations. The assessment criteria are subject to the tester’s subjective understanding and judgment, and the measurement results can be significantly influenced by subjective factors, as the subjects may conceal information in the questionnaire survey. The summary of related works is showed in Table 1.

## 3. Materials and Methods

### 3.1. Overall Process

The overall framework of this study model is shown in Figure 1. Firstly, the subjects’ facial videos were captured via a camera and saved in a .AVI format. Then, the face detection algorithm was used to obtain the facial image in each frame of the video. The physiological parameters and expression information were extracted from the selected facial images. Finally, the physiological parameters and expression information were input into the random forest classification model to predict the level of impulsive aggression of each subject.

### 3.2. Physiological Parameters

Imaging photoplethysmography (IPPG) is a non-invasive biomedical detection method which is widely used for non-contact collection of an individual’s heart rate and heart rate variability parameters [20,21]. This technology uses imaging equipment to collect video information containing the measured area, record the pulse signal that changes the light intensity caused by blood volume changes in the form of video images, and extract physiological parameters such as heart rate and blood oxygen saturation through video image processing [22,23]. This study used this method to collect the subjects’ physiological parameters, as shown in Figure 2.

Using a CCD camera with a frame rate of 100 fps, a video containing the face was obtained. The face detection algorithm was used to obtain the facial image in each frame of the video, and the tracking was performed. Next, the key point detection model was used to obtain the facial key point information in the image and select the region of interest. To avoid interference from areas such as the eyes and lips, this study selected the cheek area with relatively rich facial capillaries as the region of interest (ROI) [24], as shown in Figure 3.

The selected ROI area is indicated by the green box in the figure, and its specific position is determined by two facial key points: A (X_37_, Y_29_) and B (X_46_, Y_34_). After obtaining the sensitive area, the IPPG signal of the G channel was extracted for data preprocessing [24], including smoothing prior filtering (SPA), moving average detrending, Butterworth bandpass filtering, wavelet filtering, and other methods. Firstly, the low-frequency interference of the signal was removed using the smoothing prior method. The smoothing prior filter uses a relatively universal parameter estimation algorithm known as regularized least squares, as shown in Equation (1):(1)$${\overset{\wedge }{\theta }}_{\lambda }=\arg\underset{\theta }{\min}\left\{{\left\|H\theta -z\right\|}^{2}+\lambda^{2}{\left\|D_{d}\left(H\theta \right)\right\|}^{2}\right\}$$
where *θ* is the regression parameter, *λ* is the regularization parameter, *H* is the observation matrix, *D_d_* is the discrete approximation of the *d*-order derivative operator, and *z* is the RR interval time series, which can be expressed as:(2)$$z={\left(R_{2}-R_{1},\;\;R_{3}-R_{2},\ldots ,\;\;R_{N}-R_{N-1}\right)}^{T}$$
where *N* is the number of *R* peaks detected.

There was still high frequency noise in the signal after filtering, which exceeded the frequency of the pulse wave information. This was filtered out using moving average filtering. The formula is:(3)$$y(n)=\frac{x(n)+x(n+1)+\cdots +x(n+M-1)}{M}$$
where *x*(*n*) is the input signal, *y*(*n*) is the output signal, and *M* is the step length.

After the moving average filtering, the signal was input into a Butterworth bandpass filter (0.83–3.33 Hz) corresponding to the individual heartbeat frequency range. The signal is then subjected to multi-level Daubechies wavelet decomposition, and the high-frequency components of the 5th, 6th, and 7th layers were reconstructed to obtain the final signal. Finally, the preprocessed signal was normalized, and the pulse wave information was extracted to calculate heart rate variability (HRV) parameters.

This paper analyzes HRV information based on three aspects: time domain, frequency domain, and nonlinear domain [25,26]. The extracted HRV parameters are shown in Table 2.

### 3.3. Expression Parameters

In order to achieve accuracy and stability in facial expression recognition, this study adopted the ResNet-34 residual network model for expression recognition [27]. This is currently a commonly used facial expression recognition network. The problem of gradient disappearance and gradient explosion is solved through residual block avoidance, and at the same time, deeper features can be learned to capture complex facial emotion and expression information. The detailed structure of ResNet-34 is shown in Figure 4.

The selected dataset was the JAFFE facial expression dataset [28], which includes 7 facial expressions: 0—anger, 1—disgust, 2—fear, 3—happy, 4—sad, 5—surprise, and 6—neutral. The negative emotions are anger, disgust, fear, and sadness, while the positive emotions include happy and surprise. An increase in the proportion of overall negative emotions to the total number of emotions indicates a heightened likelihood of the individual being in a negative emotional state. Consequently, during such times, they are more susceptible to experiencing higher levels of impulsive aggression and are more prone to engaging in impulsive aggressive behavior [7]. Therefore, this study used the impulsive emotion value *E*_MO_ to quantify an individual’s facial expression information, which was calculated as follows:(4)$$E_{\mathrm{MO}}=\frac{S_{\mathrm{emo}}}{N_{\mathrm{emo}}}\times 100\%$$
where *N*_emo_ represents the total number of facial expressions in the video and *S*_emo_ represents the number of facial expressions corresponding to negative emotions in the video.

### 3.4. Random Forest Model

Based on the extracted feature parameters of heart rate variability and facial expression information, the individual’s impulsive aggression were predicted using a random forest classification model [29]. Random forest is an ensemble learning method that combines multiple decision trees to achieve accurate prediction. Its decision function can be expressed as:(5)$$H(x)=\left\{\begin{array}{ll} c_{j}, & \sum_{i=1}^{T}h_{i}^{j}(x)>0.5\sum_{k=1}^{N}\sum_{i=1}^{T}h_{i}^{k}(x)\\ \mathrm{reject}, & \mathrm{other} \end{array}\right.$$
where *x* is the sample, *H* is the final outcome, *T* is the total number of learners, *h_i_* is the learner, *N* is the total number of categories, *c_j_* is the category, $h_{i}^{j}$ denotes the output of *h_i_* on category *c_j_*, and *i*, *j*, and *k* are summation variables.

The structure of the random forest model is shown in Figure 5. The model consists of multiple decision trees, each of which is trained using Bagging ensemble learning technology. During classification, the physiological parameters and facial expression information extracted above are input as features into the random forest model. After passing through each decision tree, a classification result is obtained, and the final result is selected by voting based on the most frequently occurring result. Compared with a single decision tree, the random forest model can effectively overcome the overfitting problem of the decision tree and has better tolerance for noise and outliers.

## 4. Experimental Setup and Study Description

The experimental setup is shown in Figure 6. The light source was directed towards the participant’s face and reflected onto a color CCD industrial camera. The collected data were transmitted to a computer for processing via a data line. At the same time, a contact-based physiological parameter detection (CBPPD) device was used for synchronous detection to ensure the accuracy of the non-contact measurement results. The frame rate of the CCD industrial camera (GS3-U3-23S6C-C) was set to 100 fps with a resolution of 1024 × 1024, and the lens was a Kanda Mark M1214-MP2 industrial lens with a light intensity of 1000 lx and a color temperature of 4000 K. In order to improve the accuracy of the data collection, the participants were required to maintain a sitting position before the experiment and keep their head relatively fixed during the experiment. Figure 7 shows the actual experimental setup.

This experiment collected samples from 23 volunteers between the ages of 18 and 28, including 11 men and 12 women. All the volunteers were informed of the purpose, methods, and content of the experiment and participated with informed consent. All the volunteers were healthy and had no heart-related diseases. Two hours before the experiment, the subjects were asked to maintain a calm state, not to consume food or beverages that could excite or relax the nerves, and not to engage in intense exercise. During the experiment, 8 items related to impulsive aggression were selected from the State-Trait Anger Expression Inventory to assess the participants’ current emotional experience and impulsiveness [30]. Each item contained four options, with scores of 0, 1, 2, and 3, for a total score of 24. If the questionnaire score exceeded 12 points, the subject was considered to be in an impulsive and aggressive state.

As research suggests, behaviors such as violence and bullying have a higher likelihood of triggering negative emotions and impulsive aggression in individuals [31]. Thus, for our experiment, we deliberately selected a 5 min segment from the film and television work titled “The Glory”, which revolves around school violence and revenge, as our emotional induction material. To ensure the efficacy of the emotional induction process, we enlisted the assistance of several volunteers to watch the selected video segment and complete questionnaires. This was done to confirm the video’s ability to genuinely induce negative emotions and impulsive aggression in the subjects.

In order to ensure the effectiveness of the experiment, all the subjects watched this video content for the first time, then filled out the questionnaire honestly and without deliberate concealment to verify the accuracy and authenticity of the data. The specific information of the volunteers is shown in Table 3.

The specific experimental procedure is shown in Figure 8. First, the subjects rested for 30 min in a quiet room to keep their body in a relaxed state. After 30 min, the subjects were asked to sit still in a chair while their facial video was collected for a period of 5 min. These data served as the baseline data for the subjects’ calm state, and the subjects were also asked to complete the State-Trait Anger Expression Inventory questionnaire, which served as the score for the subjects’ emotional experience and impulsiveness in their calm state.

After a 5 min rest, the subjects began the emotional induction experiment. While watching the film, the subjects’ emotions were induced using guided language, and their facial videos were collected using a camera for a duration of 5 min. These data served as the data for the subjects’ negative emotional state. After the emotional induction experiment ended, the subjects were immediately asked to complete the State-Trait Anger Expression Inventory questionnaire again, and the score from this round was used as the score for the subjects’ emotional experience and impulsiveness in their induced emotional state. If there was a significant difference between the questionnaire results obtained after the task and those obtained in the calm state survey, it was considered successful in inducing negative emotions and impulsive aggression in that volunteer. Only the samples that generated negative emotions and impulsive aggression were used to evaluate the establishment of the model.

In order to ensure the effectiveness of the experimental method and the model we proposed, before the formal experiment, we used the same experiment and model to perform binary classification detection on the basic emotion “anger”, and the result accuracy was 93.67%. Therefore, we considered the experiment and the proposed model to be reliable in the subsequent detection and classification of impulsive aggression.

## 5. Results and Discussion

In the experiment, a total of 20 valid samples were collected after removing three invalid samples. Next, the samples were processed using a sliding window, with a window length of 3 min and a sliding step of 30 s, resulting in 100 groups of experimental data. The average values of all the data are shown in Table 4.

After obtaining the experimental characteristic parameters, we screened the characteristic parameters using an ANOVA (with a significance level of α < 0.05). The selected characteristic parameters should meet two requirements. First, there should be a clear difference between the parameter in the calm state and the impulsive state so the model can distinguish between whether the individual is in a calm state or an impulsive state. Second, the differences in parameters between different individuals in the same state should be small, so as to avoid misjudgment due to individual differences. The sum of squares between groups (SSB) is used to measure the difference or variability between different groups. When α_SSB_ < 0.05, we can consider that there is a significant difference between the calm state and the impulsive state. The sum of squares within groups (SSW) is used to measure the degree of dispersion of individual observations within a group. When α_SSW_ > 0.05, we can consider that there is not a significant difference between different individuals in the same state. The SSB and SSW of each feature are shown in Table 5.

After screening, the selected feature parameters that met the requirements included the mean HR, LF/HF, SD_2_/SD_1_, and E_MO_ parameters. Figure 9 shows the distribution of these feature parameters in the calm and impulsive states as boxplots. Compared to the calm state, when subjects were in an impulsive aggressive state, their mean HR, LF/HF, SD_2_/SD_1_, and E_MO_ increased.

The selected four HRV and facial expression parameters were used to train the impulsivity classification model. The training was performed using a random forest classification model, and a five-fold cross-validation was applied due to the small sample size. The model parameters were also controlled to prevent overfitting, and the optimal parameters were determined using a grid search method. The final parameters and the accuracy of the model classification results are shown in the Table 6 below:

After searching for model parameters, the final determined parameters are shown in the table. The model selected is the random forest classification model, with a maximum depth of 30, a minimum leaf node size of 4, a minimum sample size of 5 on each leaf node, and 10 decision trees. At this setting, the accuracy of the classification model was 89.39%, effectively categorizing whether an individual was in an impulsive aggression state and predicting their impulsive aggressiveness.

In order to verify the performance of the model, we also established impulsive aggression prediction models based on physiological parameters (containing only HRV feature parameters) and impulsive aggression prediction models based on expressions (containing only expression parameters) as comparative analyses. The accuracy of the prediction results for the three models are shown in Table 7 below.

Table 7 shows an accuracy comparison between our proposed model and the single models. It can be seen that our prediction model is more accurate than the others.

## 6. Conclusions

This study proposes a model for predicting impulsive aggression based on the fusion of physiological and facial expression information from video images. Compared with existing methods for predicting aggressive behavior, this model overcomes the shortcomings of strong subjectivity, low accuracy, and high concealment. The method utilizes imaging photoplethysmography (IPPG) and the ResNet-34 facial recognition model to extract physiological and facial expression information. It also uses a random forest classification model to predict an individual’s impulsive aggression. The experimental results show that the accuracy of the model reached 89.39%, which was higher than single models. It could effectively classify individuals in impulsive aggression states, and it predicted an individual’s impulsive aggression. This method is non-contact and non-invasive and can be used to screen and predict the tendency of impulsive aggression in special individuals in advance, making it more objective. Moreover, this method also provides important reference value for exploring new methods for predicting impulsive aggression. However, the current experiment also has the problem of a small experimental sample dataset. In the future, we hope to obtain more samples and further train the model to obtain more accurate results.

## Figures and Tables

**Figure 1 bioengineering-10-00942-f001:**
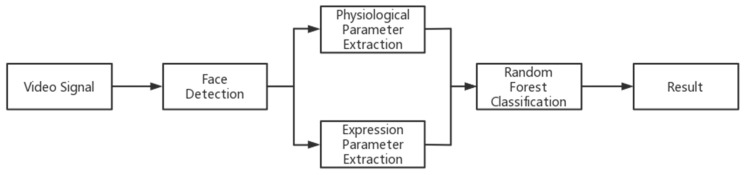
Overall flowchart. Firstly, the face video of the special individuals is captured by the camera and saved in video format. Then, the face detection algorithm is used to obtain the face image of each frame of the video. The physiological and facial parameters are extracted from the selected face images. Finally, the physiological and facial parameters are input into a random forest classification model simultaneously to obtain the impulsive aggression prediction results of the special individuals.

**Figure 2 bioengineering-10-00942-f002:**
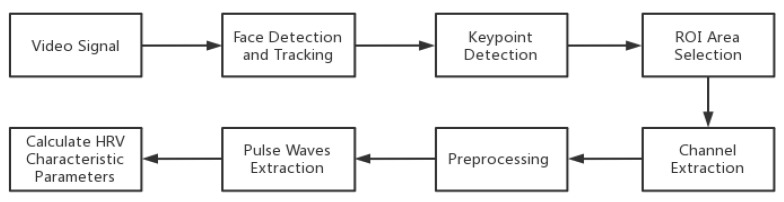
Process of non-contact extraction of physiological parameters. In order to extract the parameters, we obtained the video signal first. Then, using facial detection and tracking, along with key point detection, we successively selected the region of interest area. After that, we extracted the channel and preprocessed the signal. Finally, by extracting the pulse waves, we calculated the heart rate variability characteristic parameters.

**Figure 3 bioengineering-10-00942-f003:**
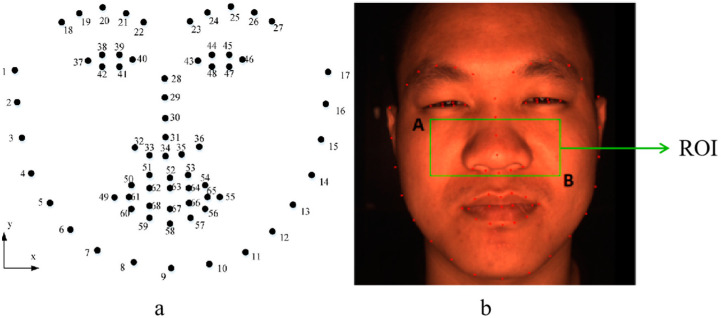
Key point detection and ROI area selection. (**a**) Schematic diagram of 68 key points on the face. (**b**) The image detected by the 68 facial features. The green box is the selected region of interest. A and B are the two vertices of the box, which are A (X_37_, Y_29_) and B (X_46_, Y_34_).

**Figure 4 bioengineering-10-00942-f004:**
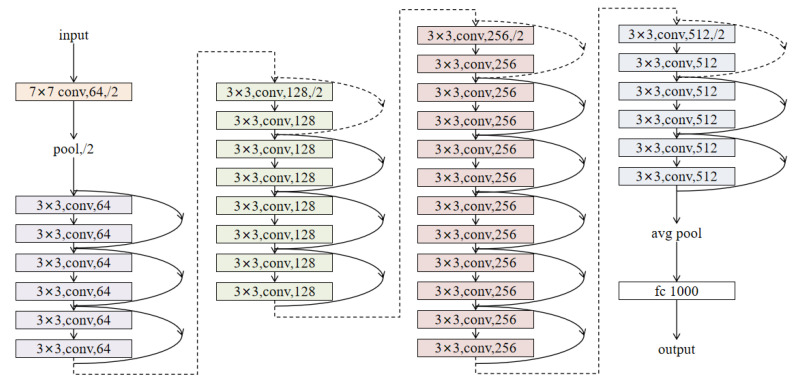
ResNet-34 residual neural network framework. The ResNet-34 residual neural network includes a convolutional layer, a pooling layer, a series of residual structures, average pooling downsampling, and a fully connected layer to obtain the final output.

**Figure 5 bioengineering-10-00942-f005:**
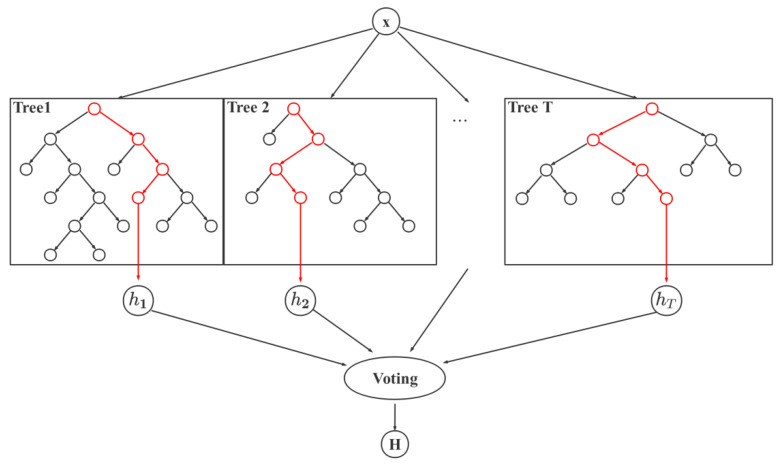
Random forest model structure. The x is the input of the model, and Tree 1 to Tree T are the different decision trees in the model. Inputting x into each decision tree will result in the classification result *h*_1_ to *h_T_*. Finally, by voting on the results of all trees, we obtain the classification result H of the model.

**Figure 6 bioengineering-10-00942-f006:**
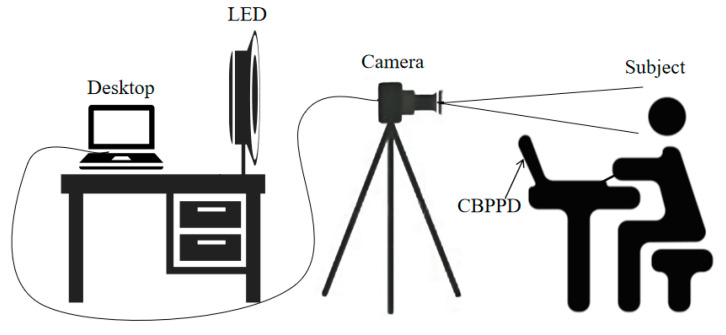
Schematic of the experimental setup. The contact-based physiological parameter detection (CBPPD) device was attached to the index finger of the participant’s left hand for PPG signal detection. The LED provided uniform illumination of 1000 lx. The camera captured the subject’s facial video image and sent the data to the desktop to calculate the result.

**Figure 7 bioengineering-10-00942-f007:**
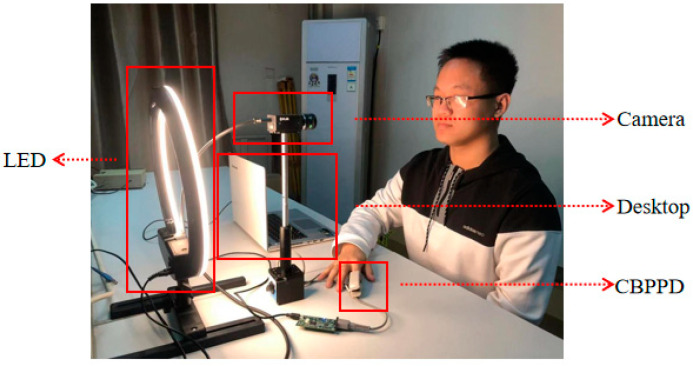
Experimental setup (dark room for actual experiments). The actual experiment was carried out in a dark room. The contact-based physiological parameter detection (CBPPD) device was attached to the index finger of the subject’s left hand for PPG signal detection. The LED provided uniform illumination of 1000 lx. The camera captured the subject’s facial video image and sent it to the desktop to calculate the result.

**Figure 8 bioengineering-10-00942-f008:**
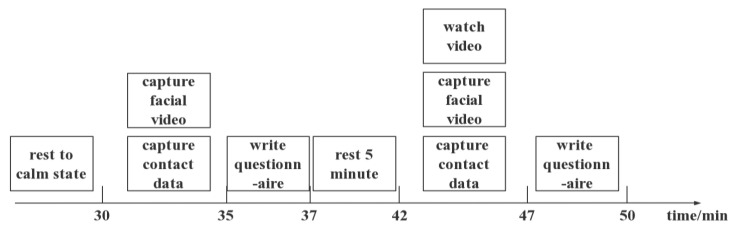
Experimental flow chart showing the experiment process. First, the subjects rested for 30 min to ensure that they were in a calm state. Then, video footage was captured when they were calm to serve as the control data, and they were asked to complete the questionnaire. Next, negative emotions were induced in the subjects, and video footage was captured simultaneously as the experimental data in the impulsive aggression state. Finally, the subjects completed the questionnaire again.

**Figure 9 bioengineering-10-00942-f009:**
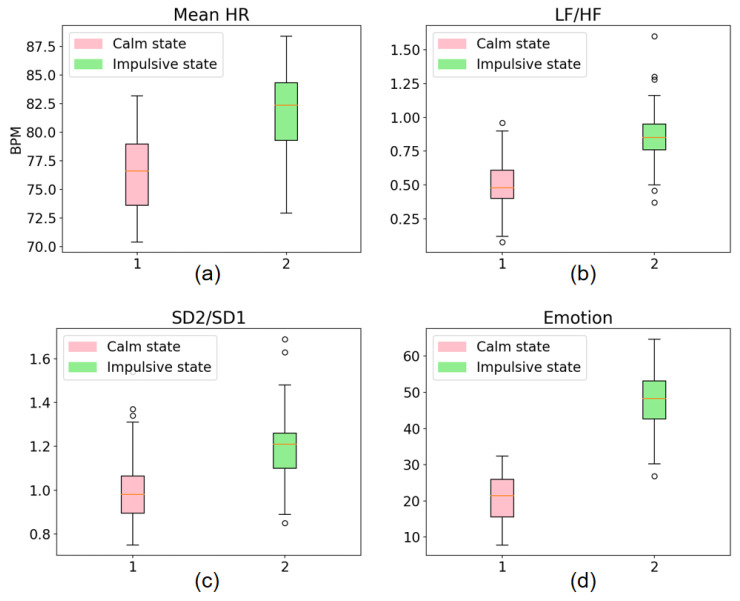
Box plot of feature parameters. The figure shows boxplots of four selected feature parameters: (**a**) mean HR, (**b**) LF/HF, (**c**) SD_2_/SD_1_, and (**d**) emotion. In each boxplot, the red color represents the distribution of data in the calm state, while the green color represents the distribution of data in the impulsive state.

**Table 1 bioengineering-10-00942-t001:** Summary of related works.

	Year	Purpose	Measures
Maria et al. [14]	2019	Trait aggression of recidivists	Questionnaire
Ricarte et al. [15]	2021	Self-injurious behaviours and suicide attempts among incarcerated people	Questionnaire
Hausam et al. [16]	2020	Scored and classified prisoners’ aggression	Behavior rating scale
Seid et al. [17]	2022	Measured antisocial personality disorder (ASPD) of incarcerated people	Face-to-face interviews and scale
Wang et al. [18]	2019	Screening for highly impulsive aggressive behavior in drug addicts	Behavioral tasks and questionnaire
Shi et al. [19]	2021	Measuring stress in drug addicts	Draw-a-Person-in-the-Rain (DAPR) test

**Table 2 bioengineering-10-00942-t002:** Heart rate variability characteristic parameters.

Index	Definition	Unit
Mean HR	Average number of heartbeats per minute	ms
PNN_50_	Heart rate of adjacent R–R intervals greater than 50 ms as a percentage of all NN intervals	%
LF	Low frequency power (0.04–0.15 Hz)	ms^2^
HF	High frequency power (0.15–0.4 Hz)	ms^2^
LF/HF	Power ratio of low frequency band to high frequency band	%
SD_1_	Standard deviation of the vertical line in the Poincare chart	ms
SD_2_	Standard deviation along the marked line in the Poincare chart	ms
SD_2_/SD_1_	Correlation dimension	—

Table 2 shows the name, definition, and unit of the Heart rate variability characteristic parameters. Here, the mean HR and PNN_50_ are the time domain parameters. LF, HF, and LF/HF are the frequency domain parameters, and SD_1_, SD_2_, and SD_2_/SD_1_ are the nonlinear domain parameters.

**Table 3 bioengineering-10-00942-t003:** Volunteer information.

Test Information	Statistical Results (Mean ± Standard Deviation)
Age (years)	22.5 ± 3.5
Gender (Male:Female)	11:12
Height (cm)	173.6 ± 14.8
Weight (kg)	60.7 ± 15.7

**Table 4 bioengineering-10-00942-t004:** Comparison of characteristic parameters.

Index	Calm State	Impulsive State
Mean HR (BPM)	76.18	82.06
PNN_50_ (%)	40.09	51.62
SDNN (ms)	92.36	91.75
RMSSD (ms)	58.93	99.58
LF/HF	0.50	0.86
SD_1_ (ms)	94.57	147.60
SD_2_ (ms)	95.71	176.66
SD_2_/SD_1_	1.01	1.19
E_MO_ (%)	20.95	47.60

The data in the table are the comparison results of one of the subject’s parameters before and after emotion induction (corresponding to a calm state and impulsive state).

**Table 5 bioengineering-10-00942-t005:** ANOVA table of feature parameters.

Index	SSB	SSW
Mean HR	6.7 × 10^−25^	0.42
PNN_50_	1.7 × 10^−22^	0.03
SDNN	0.18	0.48
RMSSD	1 × 10^−31^	0.02
LF/HF	5.9 × 10^−34^	0.16
SD_1_	1 × 10^−79^	0.04
SD_2_	1 × 10^−79^	0.04
SD_2_/SD_1_	4.2 × 10^−17^	0.37
E_MO_	1.2 × 10^−33^	0.90

The data in the table are the results of the ANOVA for each parameter. SSB represents the between-group analysis result, and SSW represents the within-group analysis result.

**Table 6 bioengineering-10-00942-t006:** Results of model training.

Model	Max Depth	Min Samples Leaf	Min Samples Split	Estimators	Accuracy/%
Random Forest	30	4	5	10	89.39

The table shows the accuracy of the random forest classification model used in the study. The selected model is the random forest classification model, with a maximum depth of 30, a minimum leaf node size of 4, a minimum sample size of 5 on each leaf node, and 10 decision trees. At this setting, the accuracy of the classification model was 89.39%.

**Table 7 bioengineering-10-00942-t007:** Comparison of the experimental results.

Model	Accuracy/%
Physiological parameter	80.65
Expression parameter	75.32
Physiological and expression parameters	89.39

The table shows that the accuracy of the model based on physiological parameters was 80.65%. The accuracy of the model based on the expression parameters was only 75.32%, while the accuracy of the model integrating physiological and expression parameters was the highest: 89.39%.

## Data Availability

The data is stored in the Baidu network disk sharing link, the data link is: https://pan.baidu.com/s/1Ejz89G9x3zN7lpWWE2mvXw, and the extraction code is: cdgj.

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
