# Peer review of "Prediction of Impulsive Aggression Based on Video Images"

_bioengineering, 2023, doi:10.3390/bioengineering10080942_

Round 1

Reviewer 1 Report

The authors present a prediction method for impulsive aggression using video images. Some points need to be addressed to improve the paper.

The introduction section is long, covering the related methods. The related works can be presented in a separate section. At the end of the introduction section, the organization of the paper needs to be presented.

Can the generated datasets in the experiments be made available publicly? The reasons for selecting the only TV show, “The Glory,” needs to be justified. Why were other videos not selected? Which parts of the videos were used for experiments? Have the participants watched the series before or they were watching it for the first time. The details need to be included so that the other researchers can regenerate the results.

The future research directions need to be included in the conclusion section.

Author Response

We would like to thanks for your efforts and time in reviewing our manuscript titled “Prediction of impulsive aggression based on video images” with manuscript number: bioengineering-2492755, and providing so many helpful comments and suggestions. We make a detailed response to the your comments, and revise the content of the manuscript carefully according to the valuable suggestions from Reviewer #1 and Reviewer #2 point by point. We have made every effort to fully address these comments in the revised version, and we hope that this revised version of our manuscript is now acceptable for publication in Bioengineering. We believe that all reviewers’ comments have helped me greatly improve this manuscript, main changes to our revised manuscript are all highlighted by using red colored text. If you have any questions, please do not hesitate to contact us. Our e-mail: 3120225336@bit.edu.cn, and our address: Beijing Institute of Technology, Beijing, China.

Thanks again for your valuable comments and suggestions. I am looking forward to your reply soon.

Yours sincerely,

Liquan Dong

Reviewer 2 Report

This paper proposes a novel method for predicting impulsive aggression by combining remote PPG technology and facial expression recognition technology. The following aspects require improvement:

- A clear definition of "impulsive aggression" as a new complex emotion is needed, utilizing existing basic emotions and Russell's emotion model.

- A more detailed explanation of the iPPG method and facial expression recognition method used in the paper is required. Additionally, evidence should be provided to demonstrate their appropriate functioning within the database used in this study.

- The sentence "Multiple studies have shown that there is a significant correlation between impulsivity and negative emotions, particularly anger, anxiety, and other negative emotions" needs to be analyzed further and supported by specific existing research.

- It is recommended to separate the existing studies into a separate section, categorize them based on appropriate criteria, and present them in a table format.

- What method was used for face landmark detection? The following sentence suggests the need for additional references: "To avoid interference from areas such as the eyes and lips, this study selects the cheek area with relatively rich facial capillaries as the region of interest (ROI, Region of Interest), as shown in Figure 3."

- The following sentence, which has important implications for the study's results, requires relevant references: "When an individual has a higher tendency towards impulsive aggression, the proportion of negative emotions in the total emotional experience increases."

- In the experiment, 8 items related to impulsive aggression were selected from the "State-Trait Anger Expression Inventory" to assess the participants' current emotional experience and impulsiveness. This statement requires a reference.

- Was the video stimulus used in the experiment appropriate for inducing "impulsive aggression"? How was it validated? Are negative emotions and "impulsive aggression" considered to be the same?

- An explanation of the process for selecting features that meet the criteria through SSB and SSW analysis is needed.

- "Impulsive aggression" can be considered as a new complex emotion. However, a comparison with the basic emotion of "anger" is necessary. What is the binary classification accuracy of anger vs. non-anger based on facial expression recognition alone? Does it fall short of the proposed method's accuracy of 89.39%?

- The classification accuracy using facial expression recognition alone and the classification accuracy using HRV parameters should be presented separately.

Author Response

(The authors gave the same response as above.)

Round 2

Reviewer 2 Report

All of raised issues were adequately addressed in the revised manuscript.